# SPARSITY BY REDUNDANCY:
# SOLVING $L_1$ WITH A SIMPLE REPARAMETRIZATION

## ABSTRACT

We identify and prove a general principle: $L_1$ sparsity can be achieved using a redundant parametrization plus $L_2$ penalty. Our results lead to a simple algorithm, *spred*, that seamlessly integrates $L_1$ regularization into any modern deep learning framework. Practically, we demonstrate (1) the efficiency of *spred* in optimizing conventional tasks such as lasso and sparse coding, (2) benchmark our method for nonlinear feature selection of six gene selection tasks, and (3) illustrate the usage of the method for achieving structured and unstructured sparsity in deep learning in an end-to-end manner. Conceptually, our result bridges the gap in understanding the inductive bias of the redundant parametrization common in deep learning and conventional statistical learning.

## 1 INTRODUCTION

In many fields, optimization of an objective function with respect to an $L_1$ constraint is of fundamental importance (Santosa & Symes, 1986; Tibshirani, 1996; Donoho, 2006; Sun et al., 2015; Candes et al., 2008). The advantage of the $L_1$ penalty is that its solutions are sparse and thus highly interpretable. While non-gradient techniques like interior point methods can be applied to solve $L_1$-regularized problems, gradient-based methods are favored by practices due to their scalability on large-scale problems and simplicity of implementation (Schmidt et al., 2007; Beck & Teboulle, 2009). However, previous algorithms are mostly problem-specific extensions of gradient descent and highly limited in the scope of applicability. It remains an important and fundamental open problem of how to optimize a general nonconvex objective with $L_1$ regularization. The foremost contribution of this work is to propose a method for solving arbitrary nonconvex objectives with $L_1$ regularization. The proposed method is simple and scalable. The proposed method does not require any special optimization algorithm: it can be solved by simple gradient descent and can be boosted by common training tricks in deep learning such as minibatch sampling or adaptive learning rates. One can even apply common second-order methods such as Newton's method or the LBFGS optimizer. The proposed method can be implemented in any standard deep learning framework with only a few lines of code and, therefore, seamlessly leverages the power of modern GPUs.

In fact, there is a large gap between $L_1$ learning and deep learning. A lot of tasks, such as feature selection, that $L_1$-based methods work well cannot be tackled by deep learning, and achieving sparsity in deep learning is almost never based on $L_1$. This gap between conventional statistics and deep learning is perhaps because there is no method that efficiently solves a general $L_1$ penalty in general nonlinear settings, not to mention incorporating such methods within the standard backpropagation-based training pipelines. Our result, crucially, bridges this gap between classical $L_1$ sparsity with the most basic deep learning practices.

The main contributions of this work are

1. Identification and proof of a general principle: $L_1$ *penalty is the same as a redundant parametrization plus weight decay, which can be easily optimized with SGD*;
2. Proposal of a simple and efficient end-to-end algorithm for optimizing $L_1$ regularized loss within any deep learning framework;
3. A principled explanation of known and discovery of unknown mechanisms in the standard deep learning practice that leads to low-rankness and sparsity.

## 2    RELATED WORKS

**L1 Penalty**. It is well-known that the $L_1$ penalty leads to a sparse solution (Wasserman, 2013). For linear models, the objectives with $L_1$ regularization are usually convex, but they are not easy to solve because the objective becomes non-differentiable exactly at the point where sparsity is achieved (namely, the origin). Previous literature often proposes special algorithms for solving the $L_1$ penalty for a specific task. For example, lasso finds a sparse weight solution for a linear regression task. The original lasso paper suggests a method based on the quadratic programming algorithms (Tibshirani, 1996). Later, algorithms such as coordinate descent (Friedman et al., 2010) and least-angle regression (LARS) (Efron et al., 2004) have been proposed as more efficient alternatives. One major problem of the coordinate descent algorithm is that it scales badly as the number of parameters increases and is difficult to parallelize. Yet another line of work advocates the iterative thresholding algorithms for solving lasso (Beck & Teboulle, 2009), but it is unclear how ISTA could be generalized to solve general nonconvex problems. This optimization problem also applies to the sparse multinomial logistic regression problem (Cawley et al., 2006), which relies on a diagonal second-order coordinate descent algorithm. Another important problem is the nonconvex $L_1$-sparse coding problem. This problem can be decomposed into two convex problems, and Lee et al. (2006) has utilized this feature to propose the sign-feature algorithm. Our work tackles the problem of $L_1$ sparsity from a completely different angle. Instead of finding an efficient algorithm for a special $L_1$ problem, we transform any $L_1$ problem into a smooth problem for which the simplest gradient descent algorithms have been found efficient.

**Redundant Parameterization**. Previously, the fact that $L_1$ penalty is equivalent to $L_2$ with redundant parametrization has been discovered in various restricted settings. Grandvalet (1998) showed that $L_1$ penalty in an arbitrary system is approximately equivalent to an adaptive quadratic penalty where each parameter comes with an independent learnable weight decay strength. Rennie & Srebro (2005) and Hastie et al. (2015) study the effect of the same parametrization on the classic matrix factorization problem and suggest its extension to different loss functions. Hoff (2017) studied this reparametrization in the context of linear and generalized linear models. Poon & Peyré (2021) and Poon & Peyré (2022) study the problem in case of a convex and lower semicontinuous loss function. A preliminary work pointed out the connection between a fully connected neural network with $L_2$ and group lasso Tibshirani (2021). All these previous works lack two essential components of our proposal: (1) the application of the theory to deep learning practice and (2) the proposal that such an equivalence means that simple SGD can solve the $L_1$ penalty.

**Sparsity in Deep Learning**. One important application of our theory is to the understanding and achieving any type of parameter sparsity in deep learning. There are two main reasons for introducing sparsity to the model. The first is that some level of sparsity often leads to better generalization performance; the second is that compressing the models can lead to more memory/computation-efficient deployment of the models (Gale et al., 2019; Blalock et al., 2020). However, none of the popular methods for sparsity in deep learning is based on the $L_1$ penalty, which is the favored method in conventional statistics. For example, pruning-based methods are the dominant strategies in deep learning (LeCun et al., 1989). However, such methods are not satisfactory from a principled perspective because the pruning part is done in separation from the training, and it is hard to understand these pruning procedures are actually optimizing.

## 3    MAIN RESULT

Consider a generic objective function $L(V_s, V_d)$ that depends on two sets of learnable parameters $V_s$ and $V_d$, where the subscript $s$ stands for "sparse" and $d$ stands for "dense". Often, we want to find a sparse set of parameters $V_s$ that minimizes $L$. The conventional way to achieve this is by minimizing the loss function with an $L_1$ penalty of strength $2\kappa$:

$$\min_{V_s, V_d} L(V_s, V_d) + 2\kappa \|V_s\|_1. \tag{1}$$

Under suitable conditions for $L$, the solutions of $L(V_s, V_d)$ will feature both (1) sparsity and (2) shrinkage of the norm of the solution $V_s$, and thus one can perform variable selection and overfitting avoidance at the same time. A primary obstacle that has prevented a scalable optimization of Eq. (1) with gradient descent algorithms is that it is non-differentiable exactly at the points where sparsity is achieved, and this optimization problem only has efficient algorithms when the loss function belongs to a restrictive set of families. See Figure 1.

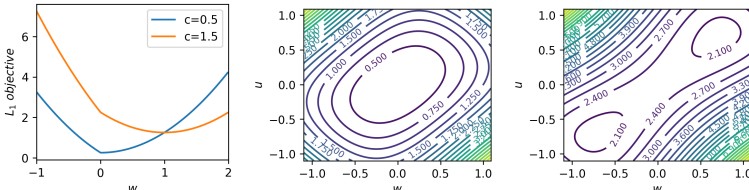

Figure 1: Landscape comparison of the original $L_1$ regularized loss and the $L_2$ regularized redundant parametrization. With the redundant parametrization, the loss becomes smooth and differentiable. The reparametrization introduces one additional minimum but is entirely benign because the two minima are identical and just mirrors of each other and converging to either achieves an equivalent performance. **Left**: the original 1d $L_1$ loss for $L = (w - c)^2 + |w|$. **Mid**: reparametrized loss with $c = 0.5$. **Right**: $c = 1.5$.

The following theorem is the first main theoretical result of this work, deriving a precise equivalence of Eq. (1) with a redundantly parameterized objective.

**Theorem 1.** *Let $\alpha\beta = \kappa^2$ and*

$$L_{sr}(U, W, V_d) := L(U \otimes W, V_d) + \alpha\|U\|^2 + \beta\|W\|^2. \tag{2}$$

*Then, $(U, W, V_d)$ is a global minimum of Eq. (2) if and only if (a) $|U_i| = |W_i|$ for all $i$ and (b) $(U \otimes W, V_d)$ is a global minimum of Eq. (1).*

The subscript $sr$ stands for "sparsity by redundancy". When $L$ is $n$-time differentiable, the objective $L_{sr}$ is also $n$-time differentiable. One can thus apply simple gradient-based optimization methods to optimize this alternative objective when $L$ itself is differentiable. When $L$ is twice-differentiable, one can also apply second-order methods for acceleration. As an example of $L$, consider the case when $L$ is a training-set-dependent loss function (such as in deep learning), and the parameters $V_s$ and $V_d$ are learnable weights of a nonlinear neural network. In this case, one can write $L_{sr}$ as

$$\frac{1}{N}\sum_{i=1}^{N}\ell(f_w(x_i), y_i) + \alpha\|U\|^2 + \beta\|W\|^2, \tag{3}$$

where $w = (U, W, V_d)$ denotes the total set of parameters we want to minimize, and $(x_i, y_i)$ are data point pairs of an empirical dataset. It is intuitive to solve this loss function with popular training methods of deep learning. The $L_2$ regularization can be implemented efficiently as weight decay as in the standard deep learning frameworks. As we discussed, this equivalence in the global minimum has been pointed out by previous works in various restricted settings (Rennie & Srebro, 2005; Hoff, 2017; Poon & Peyré, 2021). However, the equivalence in the global minimum is insufficient to motivate an application of SGD to it because gradient descent is local, and if there are many bad minima induced by this parametrization, SGD can still fail badly.

An important is thus where this redundant parametrization ind has made the optimization process more difficult or not. We now show that it *does not*, in the sense that all local minima of Eq. (2) also corresponds to local minima of the original loss and vice versa. Thus, the redundant parametrization cannot introduce new bad minima to the loss landscape.

**Theorem 2.** *$(U, W)$ is a local minimum of Eq. (2) if and only if (a) $V = U \otimes W$ is a local minimum of Eq (1) and (b) $|U_i| = |W_i|$.*

This proposition thus offers a partial theoretical explanation to our empirical observation that the optimization of Eq. (2) is no more difficult (and often much easier) than the original $L_1$-regularized loss. A corollary of this theorem is that if $L$ is convex (such as in Lasso), then every local minimum of $L_{rs}$ are global.[1] This theorem directly motivates the application of stochastic gradient descent to any problem that SGD has been demonstrated efficient for, and this theoretical justification is lacking in the previous works.

In more general scenarios, one is interested in a structured sparsity, where a group of parameters is encouraged to be sparse simultaneously. It suffices to consider the case when there is a single group because one can add $L_1$ recursively to prove the general multigroup case:

$$L(V_s, V_d) + \kappa|V_s|_2. \tag{4}$$

The following theorem gives the equivalent redundant form.

---

[1]This is reminiscent of the classical result on the deep learning loss landscape (Kawaguchi, 2016).

**Theorem 3.** *Let* $\alpha\beta = \kappa^2$ *and*

$$L_{sr}(u, W, V_d) := L(uW, V_d) + \alpha u^2 + \beta\|W\|^2. \qquad (5)$$

*Then,* $(u, W, V_d)$ *is a global minimum of Eq.* (2) *if and only if (a)* $|u| = \|W\|_2$ *for all* $i$ *and (b)* $(uW, V_d)$ *is a global minimum of Eq.* (1).

Namely, every $L_1$ group only requires one additional parameter to sparsify. Note that recursively applying Theorem 3 and setting $W$ to have dimension 1 allows us to recover Theorem 1.[2] The above theory leads to the simple algorithm given in Algorithm 1 and 2. Let $m$ be the number of groups, this algorithm adds $m$ parameters to the training process. As a consequence, it has the same complexity as the standard deep learning training algorithms such as SGD because it at most doubles the memory and computation cost of training and does not incur additional costs for inference. For the ResNet18/CIFAR10 experiment we performed, each iteration of training with *spred* takes roughly 30% more time than the standard training, much lower than the upper bound of 100%.

---

**Algorithm 1** *spred* algorithm for parameter sparsity

---

**Input**: loss function $L(V_s, V_d)$, parameter $V_s, V_d$, $L_1$ regularization strength $2\kappa$
Initialize $W$, $U$
Solve (with SGD, Adam, LBGFS, etc.) $\min_{W,U,V_d} L(U \otimes W, V_d) + \kappa(\|W\|_2^2 + \|U\|^2)$
**Output**: $V^* = uW$

---

**Algorithm 2** *spred* algorithm for structured sparsity

---

**Input**: loss function $L(V_s, V_d)$, parameter $V_s, V_d$, $L_1$ regularization strength $2\kappa$
Initialize $W$, $u$
Solve $\min_{W,u,V_d} L(uW, V_d) + \kappa(\|W\|_2^2 + u^2)$
**Output**: $V^* = uW$

---

*Two implementation caveats.* First of all, there are multiple ways to initialize the redundant parameters $W$ and $U$. One way is to initialize $W$ with, say, the Kaiming init., and $U$ to be of variance 1. The other way is to give both variables the same variance by, e.g., taking the squared root of the standard initialization methods. Secondly, even if one only wants to add $L_1$ to one layer, one should also add a small weight decay to all the other layers to prevent the model from diverging.

## 4 APPLICATIONS

In this section, we demonstrate the applications of our theory and algorithm to a series of classic tasks with both fundamental and practical importance. We first show that our algorithm is a highly scalable algorithm for solving lasso in comparisons to classical lasso algorithms such as coordinate descent and LARS. We also illustrate an application of suggested method to the sparse coding problem. Lastly, we propose a general method for performing nonlinear feature selection with neural networks (and, more generally, with ensembles of nonlinear models) and demonstrate its power for high-dimensional tasks such as gene selection.

### 4.1 A SCALABLE ALGORITHM FOR LASSO

We first test the effectiveness of the proposed algorithm for the lasso problem. We emphasize that we do not claim that the proposed method to be the state of the art method for solving lasso. In fact, it cannot be. This is because SGD is a generic optimization method that works on any differentiable problem and, by the no free lunch theorem, it certainly cannot outperform methods that are specially designed for the problem of lasso, which often take advantage of the convexity of the objective. If a SOTA performance is desired, the readers are advised to use the standard methods, such as the method proposed in Johnson & Guestrin (2015), Massias et al. (2018), or Bertrand et al. (2022). This section validates the following simple claim: *the proposed method can achieve the correct sparsity in well-understood classical problems, and does not perform badly in comparison to the default baselines of the field.*

---

[2]Note that when $L$ is a linear regression objective, the loss function is equivalent to the group lasso.

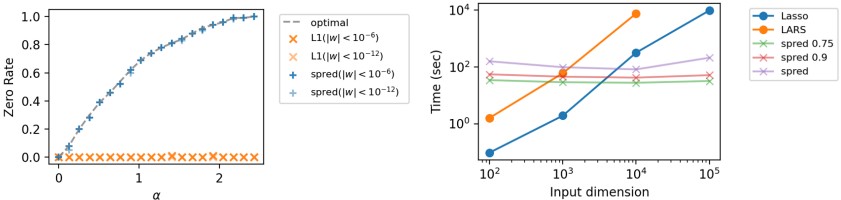

Figure 2: Performance of *spred* for solving lasso **Left**: The sparse ratio under different choices of $\alpha$. *spred* agrees with the theoretical optimum everywhere. Lasso: the closed-form solution. L1: L1 regularized least square regression solved by gradient descent; sr: the proposed method. **Right**: Up to the time scale we can tolerate, using *spred* can be 100 times faster than the conventional algorithms. Using *spred* also allows us to solve $L_1$ problems at scales at least two orders of magnitudes much larger than what the conventional algorithms can afford.

We experiment with the following two settings. *1D output and orthogonal input*: In this case, the closed-form solution for lasso is known, and this allows us to evaluate whether our method can reach the optimal lasso solutions or not. We compare with a naive gradient-descent baseline: directly applying gradient descent to the original lasso objective, denoted as *L1*. While one does not expect this method to work, it has been the popular way in deep learning to optimize the $L_1$ penalty (for example, see Han et al. (2015) and Scardapane et al. (2017)). We choose both gradient descent and Adam optimizers to optimize *spred*, as well as the original L1 regularized mean square error objective. The learning rate is chosen from $\{1, 0.1, 0.01, 0.001\}$. The final result is produced from the best setting. See Figure 2. When using *Spred*, a trick to spead up training is to set a threshold below which we set the parameter to zero at the stopping point.[3] We test two levels of threshold, and both agree with the optimal solution at convergence. We also plot in Appendix A the time evolution of the sparsity at the two thresholds, and we see that they converge to the same value at roughly the same time scale. This can be used as a criterion for assessing the convergence of *spred*. We see that *spred* agrees with the closed-form solution for all sparsity levels and for two different levels of accuracy, while the naive gradient-based method never reached a sparse solution.

*High-dimensional Lasso*. Now, we compare the optimization efficiency of *spred* with the coordinate descent and LARS solutions of lasso under different input and output dimensions. The coordinated descent solution of lasso is denoted as *Lasso*. The Least Angle Regression of lasso is denoted by *LARS*. See Figure 2-right. For *spred*, we report the time when the zero rates of the solution matrix hit 75%, 90%, and 100% of the zero rates of the converged solution. We note that there is no discernible difference in the training loss with the lasso objective for all three rates. Notably, the scaling of time complexity of *spred* is significantly better than lasso and LARS, which indicates the *spred* is particularly suitable for large-scale computation. While the time cost of running *spred* stays roughly constant as we scale to larger problems, that of the coordinate descent scales as $d$. At $d = 10^5$, using *spred* is roughly 100 times faster than the two conventional algorithms. The fact that the convergence rate of gradient descent is insensitive to the dimension in many settings is well-known in the optimization literature, which is consistent with our observation.

*Sparse coding*. As a visual demonstration, we apply the method to sparse coding (also known as dictionary learning) (Lee et al., 2006; Mairal et al., 2009). The objective for sparse coding is

$$\min_{B,S} \|X - BS\|_2^2 + \kappa\|S\|_1, \tag{6}$$

where $X \in \mathbb{R}^{d_0 \times N}$ is a set of $N$ data points, $B \in \mathbb{R}^{d_0 \times k}$ is a set of $k$ bases, and $S \in \mathbb{R}^{k \times N}$ is a set of $N$ codes that matches a subset of the bases to each data. Neuroscientifically, the bases learned in this way have been found to reproduce the Gabor filters that mimic the patterns learned in the human visual cortex (Olshausen & Field, 1997). Different from the lasso objective, this loss function is nonconvex and has spurred an interest in the optimization community to design efficient optimization methods to find solutions. One common strategy is to alternatively minimize $B$ and $S$ because when fixing the other (Lee et al.,

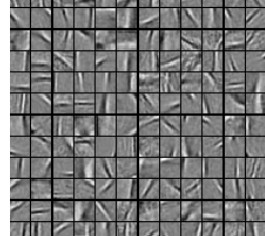

Figure 3: Examples of the learned Gabor filters from natural images.

---

[3]In experiments, we note that precise zeros can still be achieved with SGD or Adam if we train a little longer (but still within a reasonable time scale. Thus, our method, in principle, does not require a threshold.)

Table 1: Prediction accuracy for gene selection task for cancer diagnosis and survival time prediction. All tasks are classification tasks. On average, each of these datasets contains 300 data points, each with 40000 feature dimensions, and labeled into 10 classes. See Table 2 for more description.

| Dataset | HSICLasso | MLP (WD) | MLP (L1) | *spred* | |
|---|---|---|---|---|---|
| | | | | $f_l$ only | $f_l$ and $f_n$ |
| GDS1815 | $11.62 \pm 0.29$ | $0.56 \pm 0.22$ | $7.75 \pm 0.55$ | $17.75 \pm 0.77$ | $\mathbf{19.31 \pm 0.70}$ |
| GDS1816 | $13.68 \pm 0.06$ | $0.31 \pm 0.13$ | $7.12 \pm 0.75$ | $17.43 \pm 0.79$ | $\mathbf{18.75 \pm 0.77}$ |
| GDS3268 | $\mathbf{30.69 \pm 0.44}$ | $3.03 \pm 0.41$ | $15.90 \pm 0.81$ | $25.90 \pm 0.59$ | $27.86 \pm 0.65$ |
| GDS3952 | $\mathbf{45.61 \pm 0.52}$ | $14.92 \pm 1.14$ | $17.92 \pm 0.92$ | $37.00 \pm 1.22$ | $\mathbf{46.76 \pm 1.55}$ |
| GDS4761 | $42.63 \pm 0.51$ | $50.79 \pm 2.48$ | $12.62 \pm 1.78$ | $\mathbf{60.26 \pm 2.37}$ | $57.63 \pm 2.09$ |
| GDS5027 | $23.51 \pm 0.10$ | $2.55 \pm 0.48$ | $15.47 \pm 0.98$ | $\mathbf{30.37 \pm 0.97}$ | $\mathbf{30.95 \pm 0.94}$ |

2006). Here, Gabor filters can only be learned when real sparsity is achieved. Otherwise, one would get chessboard-like patterns that simple PCA would obtain. See Appendix A.3 for the experimental details.

## 4.2 NONLINEAR FEATURE SELECTION

Previously, $L_1$-based feature selection for a general nonlinear model has been an open problem because of the lack of an efficient algorithm to solve it. Existing feature-selection methods based on $L_1$ penalty are predominantly linear. The nonlinear methods are often kernel-based, where the nonlinearity comes from an unlearnable kernel. In this section, we demonstrate how the proposed method directly solves this open problem on a gene selection task (Shevade & Keerthi, 2003; Sun et al., 2015). The common gene selection tasks contain a feature dimension of order $10^4 - 10^5$ (the size of the human genome), and the number of samples (often the number of patients) is of order $10^2$. These tasks can be seen as a "transpose" of MNIST and are the direct opposite of the tasks that deep learning is good at. Additionally, one indispensable part of these tasks is that we want to not only make generalizable predictions but also pinpoint the relevant genes that have a direct physiological consequence. For example, out of roughly 50000 genes of homo sapiens, we want to know which gene is the closest associated with, say, hemophilia – such a requirement for interpretability is also challenging for deep learning. The solution to these tasks is of both scientific and practical importance. To the best of our knowledge, no deep learning method has been shown to work for these tasks (for a review, see Montesinos-López et al. (2021)). We compare with relevant baselines on 6 public cancer classification datasets based on microarray gene expression features from the Gene Expression Omnibus, including two datasets on glioma (#1815, #1816), three on breast cancer (#3952, #4761, #5027), and one on ulcerative colitis (#3268). More detailed descriptions of the datasets are in the appendix.

While a general nonlinear model can capture unexpected nonlinear effects, linear models have been found to work reasonably well for these tasks. Thus, one would like to make feature selections based on linear *and* nonlinear models. Our method allows one to achieve this goal easily: we demonstrate how to perform feature selection with an ensemble of models using *spred*. Let $f_l(W^l x)$ and $f_n(W^n x)$ denote the two different models to be trained on loss function $L(f_l, f_n)$. Here, we have explicitly written weight matrices $W^l$ and $W^n$ to emphasize that these two models start with a learnable linear layer. The following parametrization allows one to perform $L_1$ feature selection with both models:

$$\mathbb{E}_x[L(f_l(W_l(U \otimes x)), f_n(W_n(U \otimes x)))] + \kappa(\|W_l\|_2^2 + \|W_n\|_2^2 + \|U\|_2^2) \tag{7}$$

where $dim(U) = dim(x)$, and $\mathbb{E}_x$ denotes averaging over the training set. Note that the input to the two models is masked by the same vector $U$: this is crucial; without $U$, we are just training an ensemble of independent models, whereas $U$ makes them coupled. Each $U_i$ is a redundant parameter, and this is equivalent to performing $L_1$ penalty on $W_{:i}^l$ and $W_{:i}^n$ together by Theorem 3. In the experiment, we let $f_l$ be a simple linear regressor without bias and $f_n$ be a three-layer feedforward network with the ReLU activation. For simplicity, we set the objective function $L(f_l, f_n) = CE(f_l(W_l(U \otimes x), y) + CE(f_n(W_l(U \otimes x), y))$ to be the summation of two Cross Entropy (CE) losses.

See Table 1. Because the dataset size is small, for each run of each model, we randomly pick 20% samples as the test set, 20% the dev set for hyperparameter tuning, and 60% as the training set. For SGD-based models (MLP, Linear, Linear + MLP), we stop the optimization when the accuracy on

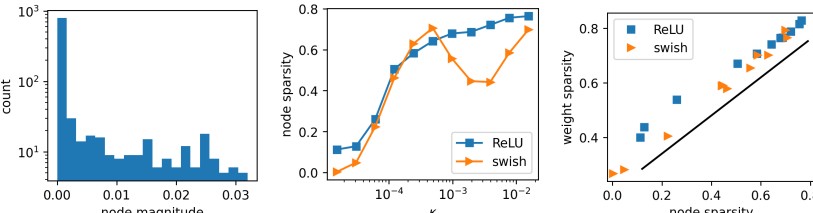

Figure 4: The feedforward structure of neural networks encourages a sparsity of the neurons (nodes). **Left**: distribution of the norm of the row vectors (each corresponding to a hidden neuron) of the last layer of a net with ReLU activation at a strong weight decay. Predominantly many nodes are either zero or close to zero. **Mid**: node sparsity of ReLU and swish as a function of weight decay strength. We see that larger weight decay leads to higher and higher neurons being constantly zero. **Right**: the weight sparsity is a linear function of the node sparsity, suggesting that the node sparsity can perfectly explain the existence of sparsity in the model.

the dev set is not increasing. The performance are averaged over 20 independent samplings of the datasets for comparison. We report the percentage of the majority class of each dataset to justify whether the models produce meaningful results. In table 1, MLP contains one hidden layer of 4096 neurons. $f_n$ contains two hidden layers of 1024 neurons. *spred* models are optimized by SGD. The learning rate and $\kappa$ are both selected from $\{7e\text{-}1, 5e\text{-}1, 3e\text{-}1, 1e\text{-}1, 5e\text{-}2, 3e\text{-}2, 1e\text{-}2\}$. Besides the deep learning methods, we also compare with HSIC-Lasso, a conventional $L_1$-based non-linear feature selection method (Yamada et al., 2014), which has been a standard method, and recent works have identified it as one of the best-performing methods for these tasks (Sun et al., 2015; Krakovska et al., 2019).

We see that deep learning combined with *spred* achieves very good performance, outperformed by the conventional method on only one dataset. On the other hand, simply applying deep learning does not work on any of the datasets. This is expected for tasks whose dimension is far larger than the available number of data points because memorization can be too easy. Importantly, simply applying $L_1$ to an MLP fails badly because gradient descent cannot find a sparse solution and thus cannot prevent overfitting. This Besides the competitive performance of the proposed method, we note that for each experiment, each training with the proposed method is at least ten times faster than the existing methods. As discussed, this is due to the dimension-free efficiency of gradient descent and expected. To the best of our knowledge, this is the first time deep learning has been shown to work for a high-dimensional feature-selection task with so few data points. Designing better architectures that suit the task of gene selection will further boost performance. Our result is thus expected to greatly accelerate the exploration and incorporation of deep learning technology into this field.

## 5 SPARSITY OF DEEP NEURAL NETWORKS

This section is devoted to applying the theory and algorithm to deep learning. We show that our theory also directly contributes to our understanding of neural networks.

### 5.1 INHERENT NODE SPARSITY IN FULLY-CONNECTED LAYERS

Our theory suggests the existence of approximate $L_1$ regularization in some systems. For example, consider a model containing a term of the form $vh(w)$, where $h(w) = aw + O(w^2)$ is a homogeneous nonlinear function. Then, in the neighborhood of zero (where sparsity is achieved), this term expands to $avw + O(w^2)$. With a weight decay present, this approximate redundancy becomes an effective $L_1$ regularization and encourages sparsity.[4] This type of function appears very frequently in the context of deep learning as two consecutive layers.

We first consider a ReLU net with neurons $784 \rightarrow 1000 \rightarrow 10$ trained on MNIST. The model can be compactly written as $f(x) = W^{(2)}\sigma(W^{(1)}x)$, where $\sigma$ is the nonlinearity and the effect of bias in incorporated into the definition of $x$. It is well-known that there is a rescaling symmetry in the

---

[4]Also note how this expansion is consistent with strong weight decay.

model. Let $h_i \coloneqq \mathrm{ReLU}(W_i^{(1)} \cdot x)$ be the $i$-th hidden neuron. Then, the model can be written as

$$f(x) = \sum_j^{1000} a_j b_j v_j^{(2)} \mathrm{ReLU}(v_j^{(1)} \cdot x) \tag{8}$$

where, for each $j$, $v_j^{(1)}$ and $v_j^{(2)}$ are normalized vectors and $a_j$ ($b_j$) are the norms of the columns (rows) original weight matrix $W^{(2)}$ ($W^{(1)}$). This means that the loss function (for example, the cross entropy) with weight decay can be written as

$$L_{\text{total}} = L(a \otimes b, v^{(1)}, v^{(2)}) + \kappa(\|a\|^2 + \|b\|^2). \tag{9}$$

Thus, our theory can be applied to show that this loss function encourages a *node sparsity* because $a$ and $b$ are element-wise redundant parametrizations of the model, on which an $L_2$ regularization is applied. Therefore, a key implication is : *ReLU layers with $L_2$ encourage $L_1$ node sparsity*. See Figure 4. We train the specified model with Adam for 50 epochs on MNIST with varying levels of $\kappa$. We measure the degree of node sparsity (operationally defined as the number of columns of $W^{(2)}$ whose average parameter magnitude is smaller than $10^{-4}$). We also measure the function of the node sparsity as a function of the weight sparsity. A clear linear relationship suggests that the weight sparsity in the model is sufficiently explained by the node sparsities caused by the weight decay. We stress that this inherent node sparsity is quite general because it is *approximately achieved* for almost any activation functions. As an example, consider the swish activation (Ramachandran et al., 2017): $\sigma_{swish}(x, W) \coloneqq Wx \otimes \mathrm{sigmoid}(\mathrm{W}x)$. When the weight decay is strong, the parameters of the model will be close to $0$, and so we can expand to the first order and extract the norm of the preceding layer: $\sigma_{swish}(x, W) \approx b\sigma_{swish}(x, v)$, and, again, there is an effective $L_1$ penalty on the norm of the weight matrices.

Note that a node becoming sparse implies that the learned representation becomes low-rank. This result offers a direct and unified explanation of the recent observed phenomenon of neural collapses in supervised learning (Xu et al., 2022; Rangamani & Banburski-Fahey, 2022; Ziyin et al., 2022), and posterior collapses in Bayesian deep learning (Wang & Ziyin, 2022), where the collapses happen when a strong effective second-order regularization (such as weight decay) is applied to the weights. This fact suggests that the regularization effect of weight decay is much stronger than previously understood. In case of node sparsity, a single redundant parameter (the norm of a row) is sufficient to regularize the whole row of the parameter and bias them towards zero. Previously, the equivalence of a full connection with $L_1$ penalty has been noticed by Neyshabur et al. (2014) and Tibshirani (2021). Our results advances these works in two regards: (1) the extension to the approximate form, which suggests that the existence of such sparsity is much more common than previously thought, (2) the proposal that this fact can be algorithmized to systematically achieve sparsity in deep learning, which we demonstrate in the next section.

## 5.2 COMPRESSION OF NEURAL NETWORKS

The proposed method also offers a principled way of performing network compression in deep learning. We experiment with unstructured weight sparsity for deep neural networks. Our method can also be used for achieving structured compression, which we leave to future work.[5] We simply apply *spred* to all the weights of a ResNet in this section. Previous methods often rely on heuristics for pruning. For example, removing the weights with the smallest magnitudes from a normally trained network. However, the problem with such methods is that one does not know in principle the effect of removing such weights, even though they seem to work empirically. Our method is equivalent to $L_1$ and has its theoretical foundation in both traditional statistics and Bayesian learning with a Laplace prior. The meaning of removing a parameter with magnitude $c$ is clear: its removal from the model will cause the training loss to increase by roughly $\kappa c$. We also emphasize that the goal of this section is not to suggest that the proposed method is a competitive method for network pruning and compression because we are not proposing a new method after all: $L_1$ is known to lead to sparsity, and our method is just a method for optimizing $L_1$ constraints. The performance of the proposed method can thus be no better than what a simple $L_1$ constraint can provide. The thesis of this section is that when an efficient way to optimize $L_1$ exists, it can perform as well as the existing methods that are not $L_1$-based, and thus $L_1$ based strategies are really worth exploring by

---

[5]For example, applying a vector of filter masks to the filters allows one to learn a sparse set of filters.

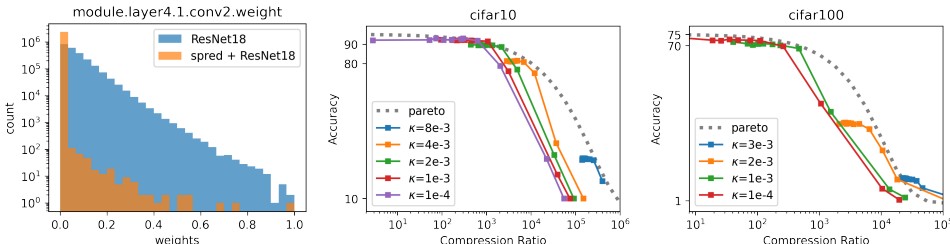

Figure 5: Performance of $L_1$-based ResNet18 pruning on CIFAR-10 and CIFAR-100. For both datasets, the estimated pareto frontier is competitive against any known existing pruning methods. **Left**: Distribution of weight parameters in the largest convolution layer of ResNet18 trained on CIFAR10. Training with *spred* leads to a very sparse distribution without affecting test accuracy. **Mid**: CIFAR-10. The grey dotted line shows the estimated pareto frontier by fitting a sigmoid. **Right**: CIFAR-100.

the community. Prior to our work, many works have attempted to naively optimize the $L_1$ constraint with SGD, but such approaches have been found to perform rather badly compared with non $L_1$-based methods (Han et al., 2015). Our result thus implies that the reason why $L_1$ has not worked well in deep learning up to today is due to a lack of a good generic optimization method, not because $L_1$ is inherently unsuitable for deep learning.

We first train a model with and without *spred* both at $\kappa = 5e - 4$ and compare the distribution of the weight. See Figure 5. Both models achieve the established accuracy of $93\%$ while the training with *spred* leads to a much sparser distribution. We now test the performance of $L_1$ for network pruning on CIFAR-10 and CIFAR-100. We resort to a simple and established procedure that has been utilized since Han et al. (2015). We first train a ResNet18 with fixed sparsity regularization for 200 epochs with SGD, prune at different thresholds, and then finetune for another 20 epochs with Adam at a learning rate of $4 - 4$. Our implementation of ResNet18 contains roughly 11M parameters, consistent with the standard implementation. See the mid (CIFAR-10) and right panels (CIFAR-100). For both datasets, the training at $\kappa = 5e - 4$ recovers the standard performance of these models. For CIFAR10, the model can be pruned up to 1000 compression ratio (*cr*, total parameter divided by the remaining parameters) without losing accuracy, consistent with the best previous results reported in (Tanaka et al., 2020). For CIFAR100, Tanaka et al. (2020) achieves $70\%$ accuracy only at 18 cr, whereas the pruning with $L_1$ achieves $70\%$ accuracy at 110 cr, roughly a magnitude better in compression power. To the more extreme end, the proposed method drops to roughly $35\%$ accuracy at 6500 cr, doubling the performance of the best-known result previously, which drops to $15\%$ at a similar 5500 cr (Tanaka et al., 2020). Our result thus promotes the use of $L_1$ penalty in deep learning. Interestingly, the higher the $\kappa$, the more suited the trained model becomes for a more aggressive pruning. $\kappa$ is thus a parameter worth finetuning if one wants to achieve the best sparsity-performance tradeoff.

We also tried using the thresholds of the trained model as a mask, which we apply to a model at initialization, and a similar performance to the finetuned model is obtained. Our result thus supports the lottery ticket hypothesis and can be used as an alternative method for obtaining a lottery ticket. In fact, the performance of the models thus obtained is much better than the results reported in the original lottery paper (Frankle & Carbin, 2018) and in a following survey (Blalock et al., 2020). We stress our thesis: $L_1$ *can indeed work in the context of deep learning if we have an efficient way to optimize it*.

## 6 Discussion

In this work, we have established a principle that a redundant parametrization can lead to an implicit constraint towards sparsity. In essence, Our theory deals with the landscape of an $L_2$-regularized redundant parametrization. Our empirical result, in turn, shows that optimization with gradient descent on such redundant but smooth landscapes is both efficient and scalable and can be of great use in various problems. We also bridged modern deep learning with conventional sparsity learning with an $L_1$ penalty, and we hope our work will stimulate more research at this intersection. For all problems we approached, we have applied $L_1$ in the most straightforward way, and developing more sophisticated methods with $L_1$ is certainly one promising future direction. A fundamental question that arises is why gradient descent can optimize such a landscape so efficiently, and future research that answers this question should advance our understanding of the optimization of deep learning models. .

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

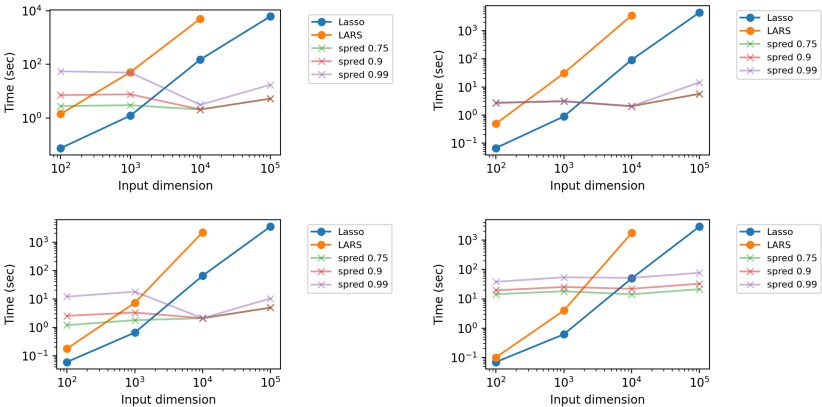

Figure 6: Performance of *spred* across different values of $\alpha$. From upper left to lower right: $\alpha = 2, 3, 4, 5$. The main text contains the case when $\alpha = 1$.

## A  EXPERIMENTAL CONCERNS

### A.1  CONVERGENCE OF SGD ON *spred* LASSO

Figure 7 presents the training trajectory under different value of $\alpha$. We can see from the left subgraph of Figure 7 that *L1* fails to produce the sparse solution even though the L1 norm of the weight matrix is almost identical to *spred*. When the $\alpha$ gets larger, our method significantly outperforms *L1* because it achieves the highest sparse ratio but almost zero L1 norm of the weight matrix.

### A.2  *spred* LASSO FOR DIFFERENT REGULARIZATION STRENGTHS

For completeness, we also compare *spred* with the baseline Lasso methods across different levels of regularization. See Figure 6. As the figure shows, our observation in the main text holds across different strengths of regularization.

### A.3  SPARSE CODING

Our theory suggests a straightforward solution using gradient descent. We parametrize $S$ by $W \otimes V$ and replace the $L_1$ norm with a corresponding weight decay for $W$ and $V$ respectively. This means that we optimize the following objective:

$$\min_{B,W,V} \|X - B(W \otimes V)\|_2^2 + \kappa(\|W\|_2^2 + \|V\|_2^2), \qquad (10)$$

This objective is differentiable and the (stochastic) gradient descent method can be applied. We normalize $B$ by its columns as in previous works. Additionally, we found that normalization $S$ by the rows significantly speeds up the training procedure. We let $k = 2048$. We train $S$ with simple stochastic gradient descent, and $B$ is trained with the LBFGS optimizer, with a batch size of $5000$. The dataset consists of $2 \times 10^5$ pieces of $16 \times 16$ patches of images taken from natural scenes. See Fig. 3 for a few examples of the found filter. The experiment finishes within 10 mins. Existing methods that are based on the lasso algorithm are found to be at least an order of magnitude slower, consistent with our finding in the previous section. Also, it is worth commenting that our method is compatible with the alternative optimization schemes and one can simply replace the Lasso step with GD on the redundant parametrization to obtain acceleration when the dimension of $S$ is high. The application of *spred* to this problem produces the correct Gabor filters as shown in Figure 3. Again, this shows that sparsity can actually be achieved with gradient descent on the suggested optimization.

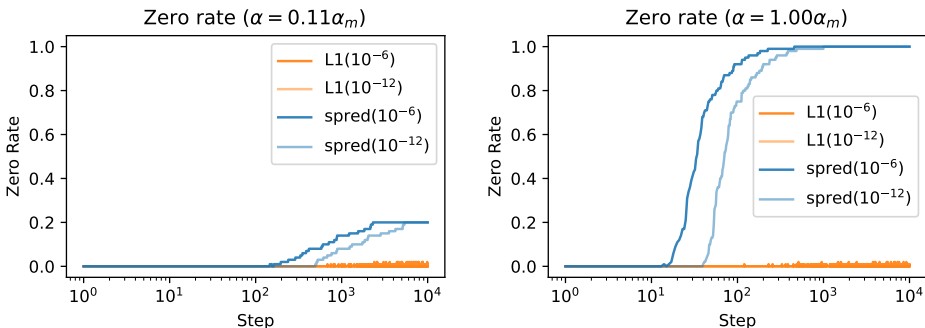

Figure 7: The training trajectory of **L1** and *spred* when $\alpha \approx 0.3$ (left) and $\alpha \approx 2.3$(right)

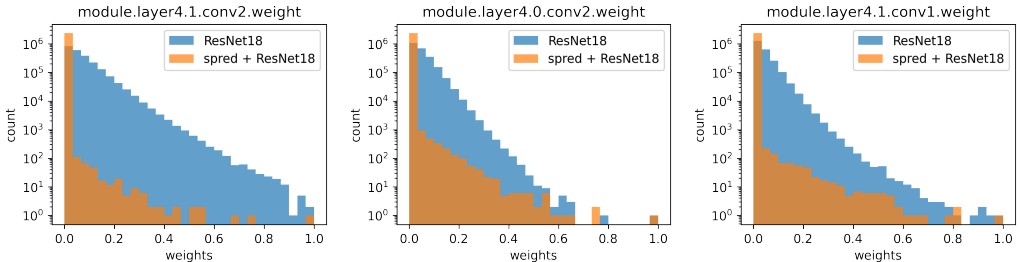

Figure 8: (normalized) Parameter distribution of the three largest convolutional layers of ResNet18 trained on CIFAR10 with SGD. The blue histogram shows the distribution of a normal ResNet18 with weight decay strength $5 \times 10^{-4}$, which is very dense. The orange shows the distribution of a *spred* ResNet18 (also with $5 \times 10^{-4}$ weight decay), which exhibits a predominant peak at zero that includes more than $99.9\%$ of all the weights parameter of the layer. This shows that training even with a very small value of regularization with *spred* already leads to a parameter distribution that favors sparsity.

Table 2: Basic statistics of seven gene datasets.

| Dataset | #features | #labels | #samples | $\frac{\#\text{samples}}{\#\text{features}}$ |
|---|---|---|---|---|
| GDS1815 (Phillips et al., 2006) | 22283 | 15 | 400 | 1.79% |
| GDS1816 (Phillips et al., 2006) | 22645 | 15 | 400 | 1.77% |
| GDS3268 (Noble et al., 2008) | 44290 | 8 | 606 | 1.37% |
| GDS3952 (LaBreche et al., 2011) | 54675 | 8 | 324 | 0.59% |
| GDS4761 (Kimbung et al., 2014) | 52378 | 7 | 91 | 0.17% |
| GDS5027 (Prat et al., 2014) | 54675 | 6 | 468 | 0.86% |

## A.4 WEIGHT DISTRIBUTION OF A TRAINED RESNET

We show more results on the weight distribution of a trained ResNet18, with roughly 11M parameters in total. We plot the parameter distribution of the three largest convolutional layers, each with roughly 2.3M parameters. See Figure 8.

## A.5 DETAILED DESCRIPTION OF THE FEATURE SELECTION TASK

See Table 2 for the statistics of the datasets. The datasets are taken from the public datasets of Gene Expression Omnibus.[6] The indices of the datasets are the same as the indices on GEO.

---

[6] https://www.ncbi.nlm.nih.gov/geo/

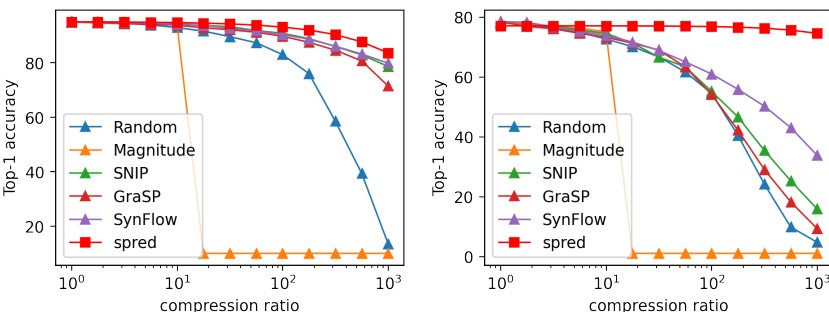

Figure 9: Performance of the proposed method in comparison to the exisitng methods that allow for compression upto at least 1000 times. **Left**: CIFAR10. **Right**: CIFAR100. This shows that $L_1$ may be a very competitive and useful baseline for model compression in deep learning.

### A.6 COMPARISON OF THE CIFAR EXPERIMENT WITH PREVIOUS RESULTS

In this section, we explicitly present the result for CIFAR10 and CIFAR100 with previous results. As we have discussed in the main text, *spred* performs very favorably in comparison with the previous results. See Figure 9. Here, we compare with the a trivial baseline of randomly pruning the network parameters. The method of pruning by magnitude after training (Gale et al., 2019; Blalock et al., 2020), SNIP (Lee et al., 2018), GRasP (Wang et al., 2020), and SynFlow (Tanaka et al., 2020). To the best of our knowledge, SynFlow is the only other method that allows one to compress a model upto $10^6$ compression ratio on CIFAR10 without becoming trivial.

## B PROOF

### B.1 PROOF OF THEOREM 1

For notational conciseness, we prove the case when $\alpha = \beta = \kappa$. The case $\alpha \neq \beta$ can be reduced to this simpler case if we redefine both $U$ and $W$ by a constant scaling. We first prove a lemma.

**Lemma 1.** *For all $i$, any local minimum of Eq. (2) satisfies*

$$|U_i| = |W_i|. \tag{11}$$

*Proof.* We prove by contradiction. Suppose not. Then there exists $U'$, $W'$ and index $i$ such that $|U_i| \neq |W_i|$ and they are a local minimum of $L(U' \otimes W') + L_2 \, reg.$, where

$$L_2 \, reg. = \kappa(\|U\|_2^2 + \|W\|_2^2). \tag{12}$$

Now, we consider an infinitesimal perturbation of the solution such that $U_i = U'_i(1 + dz)$ and $W_i = W'_i(1 - dz)$. It is straightforward to see that, by the definition of element-wise multiplication,

$$L(U' \otimes W') = L(U \otimes W). \tag{13}$$

Without loss of generality, we assume $|U_i| < |W_i|$. Now, because $U_i < W_i$, the $L_2 \, reg.$ term strictly reduces:

$$U_i'^2(1 + 2dz) + W_i'^2(1 - 2dz) - U_i^2 - W_i^2 = 2(U_i'^2 - W_i'^2)dz < 0. \tag{14}$$

This means that $U_i$ and $W_i$ cannot be a local minimum. The proof is complete. □

The above lemma implies that to find the global minimum of Eq. (2), it suffices to minimize over the solutions such that $|W_i| = |U_i|$ for all $i$. The following lemma shows that the two loss function are identical if we restrict to the domain where $|W_i| = |U_i|$.

**Lemma 2.** *Let $W \otimes U = V$ and $|W_i| = |U_i|$ for all $i$. Then,*

$$L_{rs}(W, U) = L_{L1}(V). \tag{15}$$

*Proof.* When $|W_i| = |U_i|$,

$$L_{rs} = L(U \otimes W) + 2\kappa\left(\sum_i U_i^2 + W_i^2\right) \tag{16}$$

$$= L(U \otimes W) + 2\kappa\left(\sum_i 2|U_iW_i|\right) \tag{17}$$

$$= L(U \otimes W) + 2\kappa\|U \otimes W\|_1. \tag{18}$$

By definition, $U \otimes W = V$, and so this loss is, in turn, equivalent to the following loss:

$$L(V) + 2\kappa\|V\|_1. \tag{19}$$

This finishes the proof. □

Now, we are ready to prove the main theorem. To repeat, the main theorem states the following (when $\alpha = \beta$).

**Theorem 4.** *Let* $\alpha = \beta = \kappa$ *and*

$$L_{sr}(U, W, V_d) := L(U \otimes W, V_d) + \alpha\|U\|^2 + \beta\|W\|^2. \tag{20}$$

*Then,* $(U, W, V_d)$ *is a global minimum of Eq.* (2) *if and only if* $(U \otimes W, V_d)$ *is a global minimum of Eq.* (1).

*Proof.* The theorem immediately follows from the combination of the previous two lemmas. □

### B.2 PROOF OF THEOREM 2

To repeat, the theorem states the following.

**Theorem 5.** $(U, W)$ *is a local minimum of Eq.* (2) *if and only if (a)* $V = U \otimes W$ *is a local minimum of Eq* (1) *and (b)* $|U_i| = |W_i|$.

*Proof.* For both directions, we prove by contradiction. The forward direction is much easier to prove. Let $(U, W)$ be a local minimum of $L_{sr}$ and suppose $V$ is not a local minimum of $V$. Then, one can infinitesimally perturb $V$ such that $V + dz$ has a smaller loss. This corresponds to a perturbation in $U$ and $W$ under the constraint $|U_i| = |W_i|$. By Lemma 2, $L_{rs}$ under this perturbation is also smaller than the unperturbed value. Thus, $(U, W)$ is not a local minimum – a contradiction.

We now consider the backward direction. Let $V$ be a local minimum of $L_{L1}$ and suppose $(U, W)$ is not a local minimum of $V$. As Lemma 2 shows, if we restrict to the subspace where $|U_i| = |W_i|$, there cannot be a perturbation that leads to a lower loss value because in this subspace, $L_{rs}$ is equivalent to $L(V)$. Thus, that $(U, W)$ is not a local minimum implies that there exists perturbation $dz_U$ and $dz_W$ such that $(U + dz_U, W + dz_W)$ has a smaller loss value than $(U, W)$. The loss function value is

$$L((U + dz_U) \otimes (W + dz_W)) + \kappa(\|U + dz_U\|_2^2 + \|W + dz_W\|^2) < L_{rs}(U \otimes W) \tag{21}$$

such that $|(U + dz_U)_i| \neq |(U + dz_U)_i|^2$. Now, we can construct a new parameter $U' = sign(U + dz_U)\sqrt{|(U + dz_U) \otimes (W + dz_W)|}$, $W' = (W + dz_W) \otimes (W + dz_W)$. This transformation is also infinitesimal and leaves the $L$ term unchanged. However, it strictly decreases the $L_2$ term

$$2|(U + dz_U) \otimes (W + dz_W)|^2 < \|U + dz_U\|_2^2 + \|W + dz_W\|^2. \tag{22}$$

Thus, we have constructed a model such that $|U_i'| = |W_i'|$ for all $i$, and with a strictly smaller loss. By Lemma 2, this implies that $V$ is not a local minimum of $L_{L1}$. This is a contradiction. The proof is complete. □

## C PROOF OF THEOREM 3

**Theorem 6.** *Let* $\alpha\beta = \kappa^2$ *and*

$$L_{sr}(u, W, V_d) := L(uW, V_d) + \alpha u^2 + \beta\|W\|^2. \tag{23}$$

*Then,* $(u, W, V_d)$ *is a global minimum of Eq.* (2) *if and only if (a)* $|u| = \|W\|_2$ *for all* $i$ *and (b)* $(uW, V_d)$ *is a global minimum of Eq.* (1).

The proof is similar to that of Theorem 1, and we thus only give a proof sketch.

*Proof Sketch*. When $|u| = \|W\|_2$, it is easy to check that the two loss functions agree in value. When $|u| \neq \|W\|_2$, one can always find continuous transformation (rescaling $u$ and $W$ simultaneously) of $u$ and $W$ such that the loss function is strictly reduced, and these points cannot be local minima. $\square$

The proof also shows that every minimum of $L_{rs}$ corresponds to the local minimum in the original loss, consistent with Theorem 2. This result can be immediately generalized to the case of multi-group $L_1$, where we want to apply $L_1$ (possibly with different strengths) to different groups. This can be proved by simply induction on the size of the set of groups and using Theorem 3.

