# OpenReview forum: "Sparsity by Redundancy: Solving $L_1$ with a Simple Reparametrization"
_ICLR.cc/2023/Conference — Submitted to ICLR 2023_

### Official Review · Reviewer_5KD8 · 2022-10-21

**Confidence:** 5
**Correctness:** 3
**Technical Novelty And Significance:** 1
**Empirical Novelty And Significance:** 2
**Recommendation:** 1

**Clarity, Quality, Novelty And Reproducibility:**

Unfortunately the proposed idea is not new, and the experimental investigation is not properly done


**Strength And Weaknesses:**

Strength
The application to network compression seems interesting

Weaknesses
- IMO the paper clearly lack a proper literature review: **the proposed idea, called "main results", is not new** and has been studied for a long time in the machine learning community. Recent version of this idea, usually called variational formula, can be found in [1], [2] or [3]. Authors can also read the related work of [4] for a comprehensive review on the subject.
- All the experiments **lack standard competitors**
- Section 4.1.authors compare to **gradient descent directly on the L1 loss** and LARS, which are known to not converge, or bing unstable.
I would suggest comparisons against working sets based solver such as blitz [5], celer [6] or skglm [7], and for multiple values of the regularization parameter, as it is done in [4].
- Section 4.2 simply has no quantitative results


[1] Rennie, J.D. and Srebro, N., 2005, August. Fast maximum margin matrix factorization for collaborative prediction. In Proceedings of the 22nd international conference on Machine learning (pp. 713-719).

[2] Hastie, T., Mazumder, R., Lee, J.D. and Zadeh, R., 2015. Matrix completion and low-rank SVD via fast alternating least squares. The Journal of Machine Learning Research

[3] Hoff, P.D., 2017. Lasso, fractional norm and structured sparse estimation using a Hadamard product parametrization. Computational Statistics & Data Analysis

[4] Poon, C. and Peyré, G., 2021. Smooth bilevel programming for sparse regularization. Advances in Neural Information Processing Systems

[5] Johnson, T. and Guestrin, C., 2015, June. Blitz: A principled meta-algorithm for scaling sparse optimization. In International Conference on Machine Learning (pp. 1171-1179). PMLR.

[6] Massias, M., Gramfort, A. and Salmon, J., 2018, July. Celer: a fast solver for the lasso with dual extrapolation. In International Conference on Machine Learning (pp. 3315-3324). PMLR.

[7] Bertrand, Q., Klopfenstein, Q., Bannier, P.A., Gidel, G. and Massias, M., 2022. Beyond L1: Faster and Better Sparse Models with skglm

**Summary Of The Paper:**

The paper propose a reparametrization of the L1 loss to efficiently solve L1 penalized problems.
Experiments are proposed on standard L1-penalized optimization problems and sparsity deep learning (network compression).

**Summary Of The Review:**

The application to network compression might be interesting, but the idea is not new, the paper lack a proper literature  review, experiments should be more comprehesive

---

> ### Author Response · Authors · 2022-11-19
> **author reply part 1**
>
>
> Thanks for the constructive feedbacks you provide. In response to your feedback, we now make extensive updates to discuss these relevant works to highlight why our contribution is unique, and modify the claims when no longer appropriate.
>
> **"IMO the paper clearly lack a proper literature review: the proposed idea, called "main results", is not new and has been studied for a long time in the machine learning community. Recent version of this idea, usually called variational formula, can be found in [1], [2] or [3]. Authors can also read the related work of [4] for a comprehensive review on the subject."**
>
> Thanks for point out the relevant works. We acknowledge that we neglected a series of works that are highly relevant, including the reference you give and the references the other reviewers point out. We now make extensive updates to discuss these relevant works to highlight why our contribution is unique.
>
> **"Section 4.1.authors compare to gradient descent directly on the L1 loss and LARS, which are known to not converge, or bing unstable. I would suggest comparisons against working sets based solver such as blitz [5], celer [6] or skglm [7], and for multiple values of the regularization parameter, as it is done in [4]."**
>
> This is a misunderstanding of the results in Figure 2 in Section 4.1. The right panel compares with coordinate descent algorithm, which is the standard, convergent, stable and sota method in many situations for optimizing lasso (https://hastie.su.domains/TALKS/glmnet.pdf). Gradient descent certainly does not converge, as the left panel illustrates.
>
> Also, thanks for referring to the more advanced methods for lasso, we have added them to reference. We have no doubt that these methods can outperform SGD in various scenarios. However, we emphasize that the only claim we make for the lasso example is that the suggested method is correct in the well-understood problems such as lasso and sparse coding. These sections are not meant to make methodological recommendations. We have updated draft to clarify this. This is why we do not make such comparisons at all and never made any “sota performance” claim for lasso. For completeness and better robustness, we now include additional comparisons for additional regularization strengths in section A2.
>
>
> **"Section 4.2 simply has no quantitative results"**
>
> This section is only for visual illustration, and the only claim we made in this section is that the suggested method can qualitatively capture the sparse features such as Gabor filters and is thus correct. We clarify this in the updated draft.
>
> **"All the experiments lack standard competitors"**
>
> This is not true. We had the standard and relevant competitors whenever we made a claim. As we explained above the lasso task has the most important competitor even though we do not make claims at all here.
>
> The feature selection task has the HSIC-lasso baseline, which is the most popular and standard nonlinear feature selection method in the relevant field according to the surveys we cited. This is sufficient because the only claim we make here is that “deep learning can work as well as standard methods in the realm of feature selection.”
>
> For the compression of resnet task, we used the standardized code of Tanaka et al. (2020, https://github.com/ganguli-lab/Synaptic-Flow), and, as we pointed out in the manuscript, our result is directly comparable to their results, and the other standard baselines implemented in their codebase (such as the trivial method of random pruning, the minimal method of magnitude pruning, the more advanced methods such as GRASP, SNIP, SynFlow, etc). The L1+SGD performance is shown to perform much better than the results reported in Tanaka et al. (2020).
>
> That being said, we intentionally downplayed the benchmarking aspect of the original writing because neither L1 or SGD is invented by us. Our claim is only that L1+SGD can work for deep learning, which is a major discovery in our opinion.

---

> ### Author Response · Authors · 2022-11-19
> **author reply part 2**
>
> The following 4 works are conceptually relevant to our result that we first neglected. We have added them (and quite a few other works) to the reference. We explain why our result advances those in the relevant works (**please see the rebuttal summary for a more detailed discussion and other relevant works**).
>
> [1] Rennie, J.D. and Srebro, N., 2005, August. Fast maximum margin matrix factorization for collaborative prediction. In Proceedings of the 22nd international conference on Machine learning (pp. 713-719).
>
> This work only deals with the matrix factorization problem and is a strict subclass of our main result. It also only deals with the global minimizer of the objective.
>
>
> [2] Hastie, T., Mazumder, R., Lee, J.D. and Zadeh, R., 2015. Matrix completion and low-rank SVD via fast alternating least squares. The Journal of Machine Learning Research
>
> This work only deals with the matrix factorization problem and is a subclass to which our theory and algorithm can apply to.
>
>
> [3] Hoff, P.D., 2017. Lasso, fractional norm and structured sparse estimation using a Hadamard product parametrization. Computational Statistics & Data Analysis
>
> This work makes no statement about nonlinear problems, and does not propose to solve the problem with SGD.
>
>
> [4] Poon, C. and Peyré, G., 2021. Smooth bilevel programming for sparse regularization. Advances in Neural Information Processing Systems
>
> In this work, the loss function is required to be a “convex, proper, lower semicontinuous function,” whereas our result applies to any loss function for which a global minimum exists. This work does not propose to use SGD and makes no connection to deep learning.

---

### Official Review · Reviewer_SaAB · 2022-10-24

**Confidence:** 3
**Correctness:** 4
**Technical Novelty And Significance:** 2
**Empirical Novelty And Significance:** 3
**Recommendation:** 6

**Clarity, Quality, Novelty And Reproducibility:**

- the writing style is easy to follow, though it can do better with more consistent notations and formal definitions.
- The main drawback, as discussed before, is the confusion between the perspective of the paper (modeling vs computational). The Baselines for computational comparisons (on the same problem as that of the proposed solution applies to) seem to be inadequate.
- The core contribution, Theorems 1,2,3 are quite useful for studying L1 regularization.


**Strength And Weaknesses:**

Strengths of the paper:
- The core contribution is easy to understand and has theoretical justifications. The theorems and proofs are easy to follow.
- Some interesting perspectives on sparse regularization are discussed.
--- The discussion on 5.1 giving the node sparsity perspective to weights decay is interesting.
--- The compression results in Fig 5 are impressive.

Weakness
- It is confusing if the paper is focused on computational methods or sparse models. If it is the former, the discussions on comparisons should have been more comprehensive against other computational algorithms for the same problem considered. That does not seem very conclusive. Examples below:
-- Proximal methods are popular for solving with L1 penalty. Need more clarity on why they aren't applicable for the problems considered.
-- Figure 2 is a bit unfair because it does not include any of the proximal algorithms for lasso. This is true for the sparsity plot (2a) and the time splot (2b). Need a comparison with proximal gradient methods to make any recommendations from the plots.
-- Is problem (8) equivalent to group Lasso? Are there any discussions on comparison with Group Lasso, formulation-wise and computational-wise ?

Some open questions:
- Why does the time complexity as given from Figure 2b independent of d for gradient methods ? isn't computation of the gradient itself linear in d ?
- The results of sparse coding are not given in full (Both in the main paper as well as the Appendix). The baselines are not discussed well. Neither there are any quantitative comparisons.



- (Minor) Typos / Notational inconsistencies
   -- Theorem 1 discusses w.r.t U, W, V_d whereas Theorem 2 discusses only w.r.t U, W, even though both theorems discuss the solutions for the same problem
   --  Theorem 3 is not consistent with the problem (2), rather it applies to (4).


**Summary Of The Paper:**

The paper proposes a new method for solving L1 regularized problems using redundant parametrization. Theoretical results show that the regularization on the redundant parameters is equivalent to L1 (overall, and group-wise). These results are then applied towards various use-cases ranging from simple models (Lasso) to deep NNs. Overall, the paper provides an easy way to solve with L1 penalty under various settings, though there are some aspects which need to be addressed (discussed later in this review).

**Summary Of The Review:**

Overall, the paper provides simple but useful ways to learn with L1 penalty. Different applications are discussed in the paper, though the focus might have been more towards computational comparisons.

---

> ### Author Response · Authors · 2022-11-19
> **author reply**
>
> Thanks for your constructive feedback.
>
> **"It is confusing if the paper is focused on computational methods or sparse models. If it is the former, the discussions on comparisons should have been more comprehensive against other computational algorithms for the same problem considered. That does not seem very conclusive."**
>
> We would like to point out that both readings are misunderstandings of the main claim. Whenever we try to tackle a new problem, we should always start with the simplest methods. The simplest method for optimization is clearly SGD (or GD), but this method is conventionally believed to be not working for L1 both in restricted cases and in general cases. The most important claim we make is that such SGD can actually work with L1 in general. To show this generality, we compared with the most standard and well-known methods in each relevant subject (note that SGD is much older than these methods in turn), and SGD is shown to be not bad competitors in these fields. One such example is the problem of lasso, where we compared with the most standard and sota method of coordinate descent (for example, see https://hastie.su.domains/TALKS/glmnet.pdf).
>
> **"Need a comparison with proximal gradient methods to make any recommendations from the plots."**
>
> In fact, we do not make any methodological recommendations for the lasso and sparse coding examples. The only claim we make here is that the proposed method achieved the correct and expected sparsity, unlike the naive gradient descent method. We have updated the manuscript to clarify this.
>
>
> **"Is problem (8) equivalent to group Lasso? Are there any discussions on comparison with Group Lasso, formulation-wise and computational-wise?"**
>
> “Lasso” refers to the case of a linear model. When reducing problem 8 to linear models, it is indeed group lasso. We added the relevant discussion here.
>
>
> **"Why does the time complexity as given from Figure 2b independent of d for gradient methods ? isn't computation of the gradient itself linear in d ?"**
>
> Yes, the computation of gradient is linear in d, but it is not the dominant factor when one can run gradient descent in parallel (practically, this means when the GPU still has memory). The actual dominating factor contributing to the time complexity plot is the convergence rate. This plot reflects the fact that if each gradient computation takes a constant time, what is the time it takes to reach a given loss function value, and time cost for gradient descent is known (in many cases) to be independent of d, whereas this is not true for the standard method such as coordinate GD, which cannot be run in parallel.
>
>
> **"The results of sparse coding are not given in full (Both in the main paper as well as the Appendix). The baselines are not discussed well. Neither there are any quantitative comparisons."**
>
> This example is purely for a visual illustration, and the only point we make here is that the method is working correctly. We have updated this section to clarify this.

---

### Official Review · Reviewer_D7C8 · 2022-10-24

**Confidence:** 4
**Correctness:** 4
**Technical Novelty And Significance:** 1
**Empirical Novelty And Significance:** 3
**Recommendation:** 1

**Clarity, Quality, Novelty And Reproducibility:**

The paper is clearly written.

The authors should provide the source codes for a better reproducibility.

The major concern of the paper is that the main result is known already. I refer to "Equivalences Between Sparse Models and Neural Networks" by Ryan J. Tibshirani, April 15, 2021.

**Strength And Weaknesses:**

The results are transparently presented, and the paper is quite complete and easy to follow.

The connection between deep learning methods and classical Lasso problem is extremely simple but very deep and will probably be impactful in understanding DNN performances.

The numerical experiments are also convincing.

**Summary Of The Paper:**

The paper presents a very neat result: regularization with $\ell_1$ penalty is equvalent to $\ell_{2}^{2}$ penalty up to (over) reparametrization of the model parameter. This provides several insights on the origin of sparsity in trained neural network architecture. The authors exploit this connection to provide simpler and faster solver for Lasso optimization problem.

**Summary Of The Review:**

The paper explores a nice connection between learning via overparametrized model and learning via sparsity. An explicit and fundamental correspondence between the two is established and allows for nice insights and algorithmic development for both.
Unfortunately, the main result is already published more than a year ago. I encourage the authors to review the paper I mentioned above and reconsider the contributions.

---

> ### Author Response · Authors · 2022-11-18
> **author reply**
>
> Thanks for the agreement that the insights we provided are important. We acknowledge that we neglected a series of works that are highly relevant, including the reference you give and the references the other reviewers point out. We now make extensive updates to discuss these relevant works to highlight why our contribution is unique.
>
> **"The major concern of the paper is that the main result is known already. I refer to "Equivalences Between Sparse Models and Neural Networks" by Ryan J. Tibshirani, April 15, 2021."**
>
> As we discuss in detail in the rebuttal summary, this work does not cover our result, but we agree that the essential idea is not far away. Our result is more general and deeper than this work for the following reason. This work studies the global minima of multilayer fully connected networks and showed that the global minimum coincides with that of an $L_1$ constrained version, which is not sufficient to justify the success of gradient descent on such problems. There are two major limitations of this work that we overcome: (1) the examples provided in this reference are all covered by theorem 1, which is more general because our theorem 1 can apply beyond neural networks; (2) the theoretical results only focus on the global minimum, whereas our theorem 2 provides a precise characterization of all the local minima. We have cited this work and added its discussion in the update.
>
> Besides the scientific aspect that we acknowledged above, we also point out a rule of the ICLR conference (and other major conferences): “Authors… may be excused for not knowing about papers not published in peer-reviewed conference proceedings or journals, which includes papers exclusively available on arXiv.” (https://iclr.cc/Conferences/2022/ReviewerGuide). This work you point out has not been published in any peer-reviewed venue and cannot be a basis of rejection.

---

> > ### Comment · Reviewer_D7C8 · 2022-12-09
> > **Comment on the rebuttal**
> >
> > Thanks for the rebuttal.
> >
> > However, the answer of the authors is rather strange. The authors have missed so many relevant papers on the exact same subject, that this alone justifies a direct rejection of the paper. The main result of the article is already known. And to present an already known result as a new one is a fatal flaw. The different points 1-4) mentioned seem to me rather anecdotal and often inaccurate. For example, Tibshirani's paper does not miss point 1 as the authors say.
> >
> > The authors claim that Tibshirani's "work studies the global minima of multilayer fully connected networks and showed that the global minimum coincides with that of an $L_1$ constrained version". As far as I understand, this is not correct. He exactly shows that the two problems are equivalent. The claim on the global/local minimum is a direct [important] corollary.
> >
> > The claim about non-published paper is quite fallacious. I can already write an icml article reporting a result similar to yours and present it as new? The deadline is in January and your paper will not be published in any revue at that time!
> >
> > I vote and maintain a strong reject.
> >
> >
> > Minor:
> > - introduce properly the coordinatewise product, it can be confused with Kronecker product notation.
> > - Proof of Lemma 1 is full of typo and is also unclearly written.
> > - *Write* the proof of theorem 4 ...
> > - The optimization variable $V_d$ seems rather useless here ...

---

> > > ### Author Response · Authors · 2022-12-09
> > > **Thanks for your comments**
> > >
> > > Thanks for your reply. We understand your reason for rejection, though we cannot agree with the statements in your comments.
> > >
> > > **1. "As far as I understand, this is not correct. He exactly shows that the two problems are equivalent. The claim on the global/local minimum is a direct [important] corollary."**
> > >
> > > This is not true. Lemma 1, lemma 2, and lemma 3 in the Tibshirani all deal with the global minimizer, and the equivalence is only shown for this global minimizer. You might "believe" that Tibshirani also studied local minimizes, but we think this is a false interpretation.
> > >
> > > To make the discussion more precise, we also submit three precise questions here: (a) can you quote a discussion from the Tibshirani paper that discusses "local minima"? (2) can you pinpoint which lemma/theorem in Tibshirani that directly concerns local minima? (3) Can you show the "corollary" follows from which result of Tibshirani (since such a corollary does not appear explicitly in Tibshirani)?
> > >
> > > **2. "The claim about non-published paper is quite fallacious. I can already write an icml article reporting a result similar to yours and present it as new? The deadline is in January and your paper will not be published in any revue at that time!"**
> > >
> > > We do not think it is fallacious. We are directly quoting the rule from the conference website, and why is that fallacious? You might have successfully found a potential loophole in the rule, but there is no perfect rule. In fact, we believe this rule makes a lot of sense despite the "seeming" loophole you point out.

---

> > > > ### Comment · Reviewer_D7C8 · 2022-12-09
> > > > **Comment on Desagrement**
> > > >
> > > > Thanks for your answer. Let me clarify my point.
> > > >
> > > > In Tibshirani paper, the main result is that Equation 4) with variables $u, v$ and 5) with variables $\beta$ are equivalent through the reparametrization $\beta_j = u_j v_j$. This holds for *any* $\beta, u, v$. It is a direct correspondence and so apply to global or local minimum. This is my understanding. Also note that the Tibshirani's paper does not explicitly talk about the minimizers (global or local argmin). So, if $u^\star, v^\star$ are argmin (local or global), then $\beta^\star$ defined such as $\beta_{j}^{\star} = u_{j}^{\star} v_{j}^{\star}$ is also an argmin (local or global). The objective function is invariant wrt to the parametrization.
> > > >
> > > > I am not talking about any loophole. Let's not play with words and engage in good faith. Would you, as a reviewer accept my next icml submission (taking your exact same idea) yes, or no?
> > > >
> > > > In conclusion, I believe your paper as potential, and I enjoy reading it. However, it fails on several aspect and all reviewers take serious time reading it and make relevant comment (this is not common in ML). So, please consider the comments and improve your paper.

---

> > > > > ### Author Response · Authors · 2022-12-11
> > > > > **reply**
> > > > >
> > > > > Thanks for your reply. Let us first clarify that we are also in this discussion for a good faith, and we are trying our best to improve our manuscript and to clarify what our contribution actually is, which is only possible if we receive careful and constructive feedback from the reviewers.
> > > > >
> > > > > Most importantly, it is only out of good faith that we argue our result is more advanced and general than the Tibshirani work, and it is incorrect to say that our result is already known from the Tibshirani paper. In fact, we think your understanding of the Tibshirani paper is not accurate.
> > > > >
> > > > > As you argue, the main result in Tibshirani paper is that eq. 4 is equivalent to eq. 5; this is also what the author states as his main contribution in the context. Note that Eq. 4 can be written as $\min_x A$ for a function $A$, and Eq. 5 can be written as $\min_x B$ for a different function $B$.
> > > > >
> > > > > The main result of Tibshirani is thus $\min_x A(x)= \min_x B(x)$. In all circumstances, the mathematical notation $\min$ only refers to the **global minima** of $A$ and $B$, and does not imply anything about the local minima of the $A$ or $B$. Similarly, in all circumstances, $\arg\min$ always refers to the global minima and does not imply anything about any local minimum.
> > > > >
> > > > > Also, the result that $\min_x A(x)= \min_x B(x)$ does not imply $A(x) = B(x)$. In fact, there exists plenty of $u$ and $v$ such that $A$ is not equal to $B$ at all, contrary to your understanding, which we quote here: "This holds for any $\beta, u, v$. It is a direct correspondence and so apply to a global or local minimum."
> > > > >
> > > > > Lastly, Eq. (4) and Eq. (5) (and the more general equations such as Eq. (13) and (14)) are only special cases of our general formulation and loss function because our formulation is not just limited to neural networks. We do not think this part of our result is covered by Tibshirani in the most general sense, either.
> > > > >
> > > > > Therefore, we think your conclusion that Tibshirani paper already includes our result is not correct and is due to an over-interpretation of his result.

---

> > > > > > ### Comment · Reviewer_D7C8 · 2022-12-13
> > > > > > **Re-**
> > > > > >
> > > > > > Thanks for your reply and clarifications.
> > > > > >
> > > > > > I did not state anywhere that $min_x A(x) = min_x B(x)$ implies that $A(x) = B(x)$. Which is obviously not correct, not even for the argmin. It is not difficult to find two functions with the same argmin and have different values somewhere else. My claim was that the (explicit) parametrization provides a direct correspondence between the solutions.
> > > > > > That being said, I agree with the authors that this holds for global solution and the result for local minimum is an actual (non-trivial) contribution of this paper.
> > > > > >
> > > > > > The main theorem 1 was known, the remaining novelty is the theorem 2. As the authors stated:
> > > > > > *this equivalence in the global minimum has been pointed out by previous works in various restricted settings (Rennie & Srebro, 2005; Hoff, 2017; Poon & Peyre, 2021). However, the equivalence in the global minimum is insufficient to motivate an application of SGD to it because gradient descent is local, and if there are many bad minima induced by this parametrization, SGD can still fail badly.*
> > > > > > Now, since this is the only theoretical novelty, such claims might benefit from more precise statement and justifications; specially, it seems that the theorem 2 does not say anything about the "quality" of the local minima.

---

### Official Review · Reviewer_GwGp · 2022-10-24

**Confidence:** 5
**Correctness:** 4
**Technical Novelty And Significance:** 2
**Empirical Novelty And Significance:** 1
**Recommendation:** 3

**Clarity, Quality, Novelty And Reproducibility:**

Mostly clear, a few typos.
Quality: low, for aforementioned reasons.
Novelty: probably novel.
Reproducibility: reproducible, but the authors should follow the advice of Blalock et al. (which was cited) to make sure their results can be easily compared to other shrinkage methods.

**Strength And Weaknesses:**

There has been so much work on L1 optimization over the last 15-20 years, it is difficult to keep track of it all. The proposed method is nice, in that it reduces the non-smooth objective to a smooth one that can be solved by any general method including stochastic methods. I vaguely remember seeing a method like this maybe ten years ago, certainly some methods which split the variables in two like this, and I'm a little worried that this has been discovered before. Have any other reviewers seen something similar? But I am not certain, so I won't hold it against the paper.

Another downside to working on a problem with so much prior work is that it is difficult to compare with everything. I can think of two uncited methods that have all the advantages of the proposed method without its two principle disadvantages, namely that it requires doubling the number of variables and that it does not set weights to exactly zero and therefore requires a pruning step. For smaller datasets, I would think a method like Orthant-wise L-BFGS should be at least as fast, maybe much faster. (That algorithm requires complete passes over the data, but converges in a relatively small number of steps and does produce exact sparsity.) For larger datasets, there is the method of https://arxiv.org/pdf/1505.06449v3.pdf that is stochastic, can be combined with accelerated stochastic methods like Nesterov acceleration, Adagrad/Adam, and also produces exact sparsity.

The empirical result section is lacking. In Figure 2 it is not surprising that unmodified stochastic algorithms applied to the L1 regularized objective do not produce sparsity (even *near sparsity* with parameters < 1e-6) because the non-differentiability at 0 will cause updates to hop over the value 0, unless the learning rate is extremely small. I don't understand the right half of figure 2: how is it possible that spred works equally fast on all input sizes? I hope the authors can explain that in the rebuttal. Table 1 only compares with unmodified SGD applied to the L1 objective, which as I said before should not be expected to work. The authors cite Gale et al. and Blalock et al. which discuss a wide variety of L1 algorithms-- some of these must be used as a baseline. At the very least, you have to compare to simple magnitude pruning which was shown in Gale et al. to be competitive to more complex methods. Also the benchmark tasks are small and outdated. Blalock created "shrinkbench", a modern test suite for shrinkage methods; it should be relatively easy to compare on those datasets, no?

**Summary Of The Paper:**

Another method for optimizing a loss function plus L1 penalty is proposed that splits each variable into a product and adds an L2 penalty to both terms. Experiments compare sparsity, computational efficiency and accuracy on some datasets.

**Summary Of The Review:**

A nice (in some ways) and probably novel solution to optimizing an L1 regularized objective, but the method has drawbacks (doubling the number of variables, not producing exact sparsity), several relevant methods were not even discussed (Andrew & Gao, Lipton & Elkan) and the empirical evaluation is lacking in several respects.

---

> ### Author Response · Authors · 2022-11-18
> **author reply**
>
> Thanks for your detailed and constructive feedback.
>
> **“Have any other reviewers seen something similar?”**
>
> Thanks for mentioning this. There are indeed relevant works that we neglected to cite in the original manuscript. See our discussion of these works in the rebuttal summary thread. We argue that the main contribution remains largely intact in light of all the relevant works the reviewers point out.
>
> **“It requires doubling the number of variables and that it does not set weights to exactly zero and therefore requires a pruning step.”**
>
> We would like to argue that these are not the main weaknesses of the suggested method.
> 1. We empirically demonstrated that doubling the number of variables is not too much a problem. For example, we doubled the parameters of ResNet18 from 10M to 20M, and this only leads to roughly a 10% increase in computation time, which should be acceptable to most practitioners of the field. At the same time, has any of the previous works you have in mind been tested on a problem with more than 10M parameters?
>
> 2. It is incorrect to say that our method requires a pruning. In fact, we are only suggesting the trick of pruning to speed up the optimization for practitioners. In the code we provided, exact zeros are reached without much problem and at a very reasonable time scale, as you can check by running our code. We now make this point clear in the manuscript.
>
> **"For smaller datasets, I would think a method like Orthant-wise LBFGS should be at least as fast, maybe much faster"**
>
> There are certainly a lot of methods that have been especially developed for the problem of lasso, and we never claimed to outperform those methods (and it is not our goal to). The point of the lasso and sparse coding experiment is only to show that the proposed method works correctly on well-understood examples. We now state this clearly in the main text.
>
> **"The empirical result section is lacking. In Figure 2 it is not surprising that unmodified stochastic algorithms applied to the L1 regularized objective do not produce sparsity (even near sparsity with parameters < 1e-6)"**
>
> Figure 2 left is only for the purpose of illustration for those who do not get the basic point. Figure 2 right compares with the most important methods, including coordinate descent, which has been the state-of-the-art method for lasso optimization until very recently according to. The only claim we make here is just that the method works correctly reasonably.
>
> **"I don't understand the right half of figure 2: how is it possible that spred works equally fast on all input sizes?"**
>
> Here, the reason is that the convergence speed of gradient descent, as commonly conjectured, is dimension free, meaning that to it number of iterations for GD to reach a given loss value is independent of the dimension of the parameter space (for example, see https://courses.cs.washington.edu/courses/cse547/18sp/slides/dim_free_opt.pdf). This fact is not enjoyed by the baseline methods such as lasso, which is based on coordinate descent.
>
> The computation of gradients is also linear in d, but it is not the dominant factor when one can run gradient descent in parallel (practically, this means when the GPU still has memory). The dominating factor is the convergence rate. This plot shows the following: if each gradient computation takes a constant time, what is the time it takes to reach a given loss function value? The time cost for gradient descent is known (in many cases) to be independent of d.
>
>
>
> **"The authors cite Gale et al. and Blalock et al. which discuss a wide variety of L1 algorithms-- some of these must be used as a baseline. At the very least, you have to compare to simple magnitude pruning which was shown in Gale et al. to be competitive to more complex methods. Also the benchmark tasks are small and outdated. Blalock created "shrinkbench", a modern test suite for shrinkage methods; it should be relatively easy to compare on those datasets, no?"**
>
> We did compare on a standard codebase. We were thinking about using shrinkbench, but through communication with a few authors in the field, we noted a few inconsistencies in their codes, and this is why we instead used the codebase of Tanaka et al. 2020 (see: https://github.com/ganguli-lab/Synaptic-Flow), which is actually the state-of-the-art method for a compression ratio up to 10^6 (namely, reducing a 10M-parameter resnet to roughly 10 parameters). Our result can be directly compared with those reported in Tanaka et al., which, in turn, presented a series of important benchmark methods such as the magnitude pruning method you mentioned. We now added an explicit comparison with 5 baselines in section A6.
>
> The reason why we do not explicitly present the baseline results in the table is this: we do not want to make the impression that we are proposing a completely new method and then to benchmark it. The method we propose is just L1 + SGD, neither of which was invented by us.

---

### Author Response · Authors · 2022-11-18
**rebuttal summary**

We thank all the reviewers for the extensive constructive criticism of the work. We carefully studied the feedbacks of the reviewers and made the following main updates to the manuscript (colored in orange):
1. Extensive discussion of the related works the reviewers point out, which helps us clarify the unique contribution this work makes to the community
2. Modification of many of the statement to clarify what we are actually claiming
2. Addition of computational speed experiments for the lasso example in Section A.2
3. Addition of explicit comparison of the proposed method with 5 baselines for the task of neural network compression in Section A.6



Two main criticisms of the first version of the manuscript are (1) sometimes lacking the benchmark methods – we find this criticism overly harsh and are sometimes based on claims we never made. To improve on this point, we add extensive clarification to the manuscript to clarify our claims and ensure that these claims are validated. One main example of misunderstanding is the lasso example and sparse coding example. Here, the purpose of the examples is to show that the proposed method is **correct** in the scenarios that are well-understood, **not** to establish the method as a practical SOTA method that the practitioners should rely on. We have clarified this point in the draft.

(2) The lack of a careful review of a line of highly relevant works, which leads to the criticism that the proposed contributions are not novel. We answer this part below in detail.

In the revision, we carefully reviewed all the relevant work pointed out by the reviewers, and we believe that the main contributions of the work remain largely intact because there are 4 main limitations of the previous results that we overcome. Note that the application to deep learning is justified only when all the 4 limitations are overcome:

1. previous results only focus on special cases of the equivalence (matrix factorization, convex loss, fully connected nets, etc.), whereas the proposed theory is completely general and apply to any scenario;
2. previous literature only focus on the equivalence of the global minimizer, whereas our result studied all the local minima. The second point is crucial because only having the same global minimum is unable to justify or explain the success of gradient descent on this reparametrization. Our local minima result, in sharp contrast, provides a direct and crucial justification: the reparametrization does not make the landscape more difficult than the original one.
3. Previous results lack the proposal of applying SGD to these problems and the empirical demonstration of its success
4. Lack the proposal to link to deep learning

---

> ### Author Response · Authors · 2022-11-18
> **rebuttal summary part 2**
>
> We now review the suggested references of the reviewers one by one in detail to explain why our result is novel despite these works.
>
> [1] "Equivalences Between Sparse Models and Neural Networks" by Ryan J. Tibshirani, April 15, 2021.
>
> This work studies the global minima of multilayer fully connected networks and showed that the global minimum coincides with that of an $L_1$ constrained version. This work overcame point 4 above, but has limitations 1, 2, 3.
>
> [2] Rennie, J.D. and Srebro, N., 2005, August. Fast maximum margin matrix factorization for collaborative prediction. In Proceedings of the 22nd international conference on Machine learning (pp. 713-719).
>
> This work only deals with the matrix factorization problem, and is a strict subclass of our main result. It also only deals with the global minizer of the objective. It does propose gradient descent as a solver, but has limitations 1, 2, 4
>
> [3] Hoff, P.D., 2017. Lasso, fractional norm and structured sparse estimation using a Hadamard product parametrization. Computational Statistics & Data Analysis
>
> This work makes no statement about nonlinear problems, and does not propose to solve the problem with SGD. This work has limitations 1, 3, 4
>
> [4] Hastie, T., Mazumder, R., Lee, J.D. and Zadeh, R., 2015. Matrix completion and low-rank SVD via fast alternating least squares. The Journal of Machine Learning Research
>
> This work only deals with the matrix factorization problem and is a subclass to which our theory and algorithm can apply to. It has limitations 1, 2, 3, 4
>
> [5] Poon, C. and Peyré, G., 2021. Smooth bilevel programming for sparse regularization. Advances in Neural Information Processing Systems
>
> In this work, the loss function is required to be a “convex, proper, lower semicontinuous function,” whereas our result applies to any loss function for which a global minimum exists. This work is lacking on point 1, 2, 4.
>
>
> Closely studying the references the reviewers provided, we also noticed the following relevant works that we also cite and discuss in the update:
>
> [6] Poon, C., & Peyré, G. (2022). Smooth over-parameterized solvers for non-smooth structured optimization. arXiv preprint arXiv:2205.01385.
>
> As in [5], this work only deals with a “convex, proper, lower semicontinuous function.”
>
> [7] Neyshabur, B., Tomioka, R., & Srebro, N. (2014). In search of the real inductive bias: On the role of implicit regularization in deep learning. arXiv preprint arXiv:1412.6614.
>
> This work is the one of the first works to point out the equivalence to group lasso in case of a fully connected neural network. However, it lacks 1, 2, 3
>
> We have rewritten the related works section in light of the related works the reviewers point out. We also discuss the relevance to these works when appropriate in the main results section.

---

### Decision · Program_Chairs · 2023-01-20

**Decision:**

Reject

**Justification For Why Not Higher Score:**

The lack of novelty and the poor experiments comparisons are the main reasons.

**Justification For Why Not Lower Score:**

The score is already quite low. Some experimental proposition seems to be of interests to some reviewers though (network compression).

**Metareview: Summary, Strengths And Weaknesses:**

The overall contribution is to use a simple method for solving general L1 penalized objectives with gradient descent, using a redundant l2 penalty.
The reviewers agree that the idea proposed is not new, that the paper lack a proper literature review and that experiments were limited.
Hence I follow their choice to reject the paper.